# Upper Critical Temperature of Iberian Pigs

**DOI:** 10.3390/ani15101374

**Published:** 2025-05-09

**Authors:** Manuel Lachica, Andreea Román, Ignacio Fernández-Fígares, Rosa Nieto

**Affiliations:** Department of Nutrition and Sustainable Animal Production, Estación Experimental del Zaidín, Consejo Superior de Investigaciones Científicas, CSIC, San Miguel 101, 18100 Armilla, Granada, Spain; a.roman@eez.csic.es (A.R.); ignacio.fernandez-figares@eez.csic.es (I.F.-F.); rosa.nieto@eez.csic.es (R.N.)

**Keywords:** breathing rate, heart rate, heat production, pig, rectal temperature, skin temperature, retained energy, respiratory quotient, upper critical temperature, voluntary feed intake

## Abstract

Pigs are sensitive to heat, producing large economic losses in pig production due to a lack of welfare. Iberian pigs (*Sus mediterraneus*) are a rustic breed that thrives in the Mediterranean forests in the southwest of the Iberian Peninsula where temperatures can be very hot. There are a range of temperatures where animals feel comfortable, known as the thermoneutral zone (or thermoneutrality). Into this zone, heat production reaches a minimal value, and it is constant and independent of the ambient temperature. This zone is delimited by the lower critical temperature and the upper critical temperature which are, respectively, the lowest and highest temperatures at which the heat loss of animals is minimal and the retained energy reaches its maximum. Cold stress (at least moderate) is well dealt with by most homeotherm animals, but heat stress is not. The aim of this study was to determinate the upper critical temperature of Iberian pigs.

## 1. Introduction

The Iberian pig (*Sus mediterraneus*) is a native rustic breed from the southwest of the Iberian Peninsula (Spain and Portugal). Although nowadays it is also frequently used in intensive production, traditionally, the Iberian pig was raised extensively in the Mediterranean forests—named dehesa—based on the availability of feed resources. There is a growing interest in semi- and extensive production systems where an efficient use of natural feed resources constitutes a key factor for sustainability and animal welfare. Nowadays, to improve productivity, a semi-extensive system where pigs are free on the land and balanced mixed diets are offered is used. This is followed by a finishing phase—named montanera—under solely extensive conditions up to a 160 kg body weight (BW) consuming acorns from oak trees (*Quercus ilex rotundifolia* and *Quercus suber*) and pasture when available (from 6 to 10 and from 1 to 1.5 kg daily [1]). The montanera phase complies with animal welfare demands and minimizes environmental degradation. However, these producing areas are under a hot summer Mediterranean climate (https://en.climate-data.org/europe/spain-5/ (accessed on 24 July 2024)) where high ambient temperatures are common during the summer (July–August). The average maximum temperatures in July range from 32 to 36 °C. Heat stress is a challenge in pig production systems, resulting in large production losses [2]. The increased temperature and more frequent heat waves in the Mediterranean area produce heat stress in Iberian pigs that affects growth, organ weights and biochemical parameters [3].

There is a range of temperatures in which animals feel comfortable, known as the thermoneutral zone (or thermoneutrality). Into this zone, the heat production (HP) reaches a minimum; it is constant and independent of the ambient temperature. This zone is delimited by the lower critical temperature and the upper critical temperature (UCT). The lower critical temperature and UCT are, respectively, the lowest and highest temperatures at which the heat loss of animals is minimal and the retained energy (RE) reaches a maximum. Both critical temperatures can vary depending on the size of the animal, feed intake, breed and environmental factors. When the temperature is below the lower critical temperature or above the UCT, the animal starts to feel stress. Cold stress (at least moderate) is easily handled by most homeotherm animals, but heat stress is not. Above the UCT, animals will actively increase heat loss to maintain their core body temperature. It should also be noted that between the upper limit of the thermoneutral zone and the UCT, animals already make adaptations to increase heat loss. From a productivity point of view, the most important effect is the reduction in RE (i.e., growth rate) due to a decrease in voluntary feed intake (VFI) to minimize the overall HP, and the heat increment of feeding in particular. Other physiological effects that reduce heat load by increasing the heat loss are enhanced sweating, breathing rate (BR), skin (ST) and rectal (RT) temperature, and heart rate (HR).

Pigs are very sensitive to heat due to their elevated basal metabolic HP and rapid growth, activating thermoregulatory responses [4]. The active changes in pigs to increase heat loss are physiological (e.g., increase in BR) and/or behavioral (e.g., lying on mud instead of soil). The UCT can change depending on age, BW and physiological stage [5,6].

It is important to know the UCT not only to retain productivity (profitability), but also because animal welfare regulations are becoming more intense for pig production. Pig welfare can be improved by following a priori simple measures (for example, reducing the feed ration before heat stress might occur, preventing arousal in the house during the hottest period by adapting feeding times, maximizing the ventilation rate, etc.). The pig is special among mammals because it has a very limited number of sweat glands, and also has great insulation due to its subcutaneous fat depth [7]. These factors can make pigs particularly sensitive to heat stress, particularly the Iberian pig, as although it is considered to be perfectly adapted to its environment [8], the thick subcutaneous fat layer [9]—that isolates the body to avoid heat dissipation—could make it very sensitive to elevated temperatures compared to lean breeds. It was reported that compared to improved breeds, Iberian pigs have different metabolic [10] and nutritional features [11]. Unlike for improved breeds, there is no information regarding the thermoneutral zone of Iberian pigs. The lack of information on this subject leads to extrapolating values from improved breeds to Iberian pigs or to relying on assumptions about the thermoneutral zone of Iberian pigs, ignoring differences in physiological or anatomical features. Hence, these extrapolations may be misleading. Ideally, the heat tolerance for each breed ought to be determined using heat chambers [12] where HP is directly determined. Unfortunately, there are only a limited number of reliable values available in pigs to define the thermoneutral zone, and hence, the UCT.

The main objective of the present study was to obtain the UCT of Iberian pig by determining the HP, body temperature (skin (ST) and rectum (RT)) and BR. In addition, VFI, RE, respiratory quotient (RQ) and HR were also determined.

## 2. Materials and Methods

### 2.1. Animals and Diets

The experimental procedures and animal care were in agreement with the Spanish Ministry of Agriculture guidelines (RD53/2013). The Bioethical Committee of CSIC (Spanish Council for Scientific Research, Spain) and the competent authority (Consejería de Agricultura, Pesca y Desarrollo Sostenible, Junta de Andalucía, Spain, project reference 21/03/2022/045) approved all of the experimental procedures with animals used in the present study.

Eight pure Iberian barrows (Sánchez Romero Carvajal strain; Sanchez Romero Carvajal Jabugo S.A. (Puerto de Santa María, Cádiz, Spain)) of a similar age (197 ± 1.4 days) and initial BW (61.1 ± 0.86 kg) were used. The pigs were individually housed in adjustable surface slatted pens, allowing visual contact among them, and with free access to water at all times. Experimental conditions were maintained throughout the study. The diet composition is displayed in Table 1.

The diet was barley–soybean meal-based (136 g crude protein/kg and 13.1 MJ metabolizable energy (ME)/kg) with an optimal amino acid profile [14] and covered all nutrient requirements [11]. The pigs were fed once a day at 2 × ME for maintenance (422 kJ/kg^0.75^ BW [11]) at 10:30 h.

During the pre-experimental period, the pigs were allocated in an environmentally controlled room at 28 °C. The temperature was controlled using an air conditioning apparatus (LG UM36, LG Electronics Inc., Changwon, Republic of Korea). The temperature and relative humidity were recorded every 15 min by a data logger (HOBO UX100-011; Onset Computer Corporation, Bourne, MA, USA). The photoperiod was fixed to 12 h of light (08:00 to 20:00 h) and 12 h of darkness. Four days before starting the experimental period, the barrows were fed ad libitum. To avoid any stress to the pigs, they were adapted to close contact with the staff involved with handling and to spending time in the respirometry chambers.

### 2.2. Heat Production (HP) Measurements

On the 114th day from arriving at the lab, the pigs (99.1 ± 1.71 kg BW) were moved from their own slatted pen to one of the two open-circuit-type respirometry chambers (Figure 1) following the schedule displayed in Figure 2.

This schedule allowed for the equalization of all measurements, making the average age and BW identical and very similar, respectively, along all ambient temperature assays.

The photoperiod was identical to that used before. The respirometry chamber temperature was raised or diminished by 2 °C every two days; the first day for adaptation and the second for measuring (Figure 2). The day of temperature change was considered as a transition day for adaptation to the new environmental conditions [15]. The BW was registered at the beginning and end of the respirometry chamber cycle, and the BW for each temperature was interpolated. Every morning, at around 10:30 h, the chamber was opened for cleaning and to provide feed to the pigs. This process took less than 15 min. Feces and urine were removed from the slatted pen. Urine was collected into plastic buckets placed under the slatted pen containing 50 mL of 10% sulfuric acid to avoid ammonia release and consequent discomfort for the pigs. The pigs were fed ad libitum, and leftovers were removed and quantified. A 20 L volume water tank inside the chambers was refilled with fresh water. All data recordings were performed on the second day just before the temperature shift. The VFI (ME intake) was measured and the total HP determined over the previous 24 h from O_2_ consumption, and CO_2_ and CH_4_ production according to Brouwer [16]. The RQ was determined as the CO_2_ produced/O_2_ consumed ratio. The periodic calibration of the whole system was performed by injecting pure gas CO_2_ (99.998%) and O_2_-free N_2_ (to produce an O_2_ decrement) into the chamber and the calibration factors obtained were used for correcting CO_2_ production and O_2_ consumption, respectively.

### 2.3. Physiological Measures

Physiological parameters were determined immediately before the change in temperature of the chamber. The BR (breaths/min) was measured by using a stopwatch and counting, through the chamber window, the number of uninterrupted flank movements per minute. Then, the chamber was opened and RT (°C) measured using a digital thermometer inserted into the rectum; ST (°C) by means of an infrared thermometer (KODYEE Model CF-818; Kedy Tech (Ganzhou) Electronics Co., Ltd., Ganzhou City, China) always situated on the same body site (between both scapula); and HR (beats/min) using a portable veterinary pulse oximeter (UT100V Model; UTECH Co., Ltd., Chongqing, China) for determing pulse rate by fixing the device on the ears of the animal.

The temperature humidity index (THI) was calculated each day according to Rauw et al. [17], following the equation THI = 0.8 × T + RH × (T − 14.4) + 46.4;
where T = temperature in °C and RH = relative humidity expressed as a proportion.

### 2.4. Chemical Analysis

The chemical analysis of the diet is displayed in Table 1. Samples of feeds were pooled through the experiment and analyzed in triplicate for dry matter (no. 934.01), ash (no. 942.05) and ether extracts (no. 920.39) by standard procedures [18]. Total N was determined according to the Dumas method by total combustion using the TruSpec CN equipment (Leco Corporation, St. Joseph, MI, USA), and crude protein was calculated as total N × 6.25. Crude fiber (no. 978.10) was determined using an ANKOM220 Fibre Analyser Unit (ANKOM Technology Corporation, Macedon, NY, USA).

### 2.5. Statistical Analysis

The number of animals (*n* = 8) was calculated using the G*Power software (version 3.1.9.7, Heinrich-Heine-Universität Düsseldorf [19]), accepting an alpha risk of 0.05 and a beta risk of 0.2 in a two-sided test. Experimental data were analyzed by using repeated-measures analysis of variance with the animal as an experimental unit to determine the effect of temperature. The Tukey multiple range test was used to ascertain the statistical significance of differences. One-way analysis of variance (ANOVA) using the GLM procedure and a computer software package (Statgraphics Plus for Windows Version 2.0, Manugistics Inc., Rockville, MD, USA) was used to establish differences in the parameter analyzed. Differences were considered significant at *p* < 0.05.

Regressions between different metabolic and physiological parameters (Y) and ambient temperature (X) were made using the following approach:Y = a (±SE) + b (±SE) X

## 3. Results

Average temperature, humidity and THI in the environmentally controlled room were 28.2 ± 0.03 °C, 50.6 ± 0.22% and 75.9 ± 0.06, respectively. The average temperature, humidity and THI at the imposed temperatures in the respirometry chambers are displayed in Table 2.

No differences (*p* > 0.05) were found in BW, HR and RT (101.6 kg, 97.7 beats/min and 39.7 °C on average, respectively) between the ambient temperatures set at the assays (Table 3). The BR and ST progressively increased (272 and 2.4%, respectively; *p* < 0.05) along with the temperature from 24 to 32 °C, whereas VFI, ME intake, RE and RQ progressively decreased (40.3, 40.9, 65.8 and 10.5%, respectively; *p* < 0.001). The HP slightly increased at 26 vs. 24 °C but followed the same trend as the previously mentioned parameters, decreasing when the temperature increased from 24 to 32 °C (19.2%; *p* < 0.01). The ratio of RE/ME intake (gross efficiency) and HP/ME intake (gross inefficiency) decreased (39.1%; *p* < 0.01) and increased (37.7%; *p* < 0.01), respectively, with the temperature, with a marked change from 28 °C.

Linear regressions were established for VFI or ME intake (g dry matter/kg^0.75^ BW and day), BR (breaths/min), ST (°C), HP (kJ/kg^0.75^ BW and day), RE (kJ/kg^0.75^ BW and day) and RQ vs. temperature (°C):VFI = 239.1 (±23.68) − 5.58 (±0.843) × Temperature       *n* = 40; *r* = −0.74; RSD = 14.85(1)BR = −205.6 (±38.88) + 9.95 (±1.385) × Temperature      *n* = 40; *r* = 0.76; RSD = 24.71(2)ST = 34.2 (±0.88) + 0.12 (±0.031) × Temperature          *n* = 40; *r* = 0.53; RSD = 0.559(3)HP = 1187.5 (±126.83) − 17.29 (±4.507) × Temperature    *n* = 40; *r* = −0.53; RSD = 79.89(4)RE = 2268.0 (±298.78) − 63.40 (±10.637) × Temperature   *n* = 40; *r* = −0.70; RSD = 187.3(5)RQ = 1.54 (±0.072) − 0.016 (±0.0026) × Temperature      *n* = 40; *r* = −0.71; RSD = 0.045(6)

## 4. Discussion

The world global increment of temperature represents a problem for animal production. However, studies on the heat effects on autochthonous livestock breeds are scarce. The Iberian pig is a rustic indigenous obese breed with specific physiological features (i.e., a lower protein deposition and growth rate and a higher lipid deposition than modern breeds [11]). Although the Iberian pig is well adapted to the environmental conditions, spending time in open-range grazing even under intense sunshine and elevated temperatures, no data are available in the literature about their UCT. Although the thermoneutral zone of pigs depends on various factors, it could be established between 18 and 25 °C, with temperatures above 25 °C activating thermoregulatory responses [4]. Depending on the parameter studied, the thermoneutral zone can be established with different amplitudes and therefore a variable UCT can be obtained [20]. In the present study, quadratic polynomial regression equations were performed, but the quadratic polynomial terms were not significant. According to the equations, the different plateaus (estimated UCT) were established in all cases outside the range of temperatures studied, and in some cases, with no biological meaning. For example, for RE the plateau would be reached at −28.7 °C; for BR at 36.4 °C, in contrast with our experience in previous works and also with the direct observations in the laboratory or commercial pig farms, where the Iberian pigs increased their rate at about 27–29 °C; for VFI, the plateau was at 16.7 °C, a temperature at which pigs should theoretically ingest more energy to increase their HP to maintain their body temperature. These values indicate that lower and/or higher temperatures, depending on the parameter studied, should have been included in the experimental design, although this would have made it extremely complicated, including possible animal welfare associated issues. The linear equations were chosen for their simplicity when it comes to being applied, from a practical point of view, into a similar temperature range.

In a previous study [21], no differences (*p* > 0.1) were found in HP between Iberian pigs (63 kg BW) at 20 vs. 30 °C, but the RQ value was lower for the higher temperature, indicating a reduction in lipogenesis, which made us suspect that the UCT was over the range described for modern breeds. In the present study, the UCT was in the range of the average temperatures in the hottest months in the main Iberian pig-producing areas (southwest of Iberian Peninsula) where average highs above 30 °C are common. In addition, the average BW of pigs in the present study matched with the regular BW at the end of summer when pigs are ready to initiate the finishing phase (montanera) under solely extensive conditions.

When the humidity is elevated, a relatively low temperature can produce heat stress in the animals [22]; this is why the THI should be utilized. As the THI increases, the more difficult it is to lose body heat. The maximum value for THI is 100% and this is used to assess the temperature–humidity combination in situations with heat stress risks. However, in non-sweating animals like pigs, the temperature is the most important determinant for heat stress [20]. The THI charts that appear in the literature are very variable since the size, physiological situation, etc., can be very different. For comparative purposes, in the present study, the THI chart selected was for grow–finish pigs [23] and three categories were recognized: heat stress alert, danger and emergency. Thus, in the present study, the THI ranged from the thermoneutral zone, heat stress alert, danger and emergency for 24, 26, 28 and 30, and 32 °C ambient temperatures, respectively. However, only when temperature was around 28–30 °C there was a clear effect (*p* < 0.05) on seven (VFI, BR, ST, ME intake, HP, RE and RQ) out of the nine parameters measured (Table 3).

A reduction in VFI has been reported for different pig breeds under heat stress regardless of their size or physiological state. When pigs are exposed to heat stress, feed intake (ME intake) is reduced [24,25] to decrease HP and, consequently, the amount of nutrients available for growth is diminished [26]. Lachica et al. [21] reported that, unlike improved breeds, Iberian pig ME intake (measured during one day in a respirometry chamber under long-term constant heat stress conditions) was not reduced (*p* > 0.1) at 30 °C and, therefore, growth (RE) was not affected, which may have been due to a genetic adaptation to harsh environments [8]. However, Pardo et al. [3] reported a reduction in VFI obtained over 28 experimental days in growing Iberian pigs under long-term heat stress conditions, but to a lesser extent than in lean pigs. Huynh et al. [20] pointed out that above the UCT, the VFI and HP decreased, and BR and RT increased. These authors mentioned that the UCT can be considered to be the inflection point above which VFI decreases and RT increases. Several studies reported that at high temperatures, a decrease in HP is found in relation to increasing RT [27,28,29]. Nevertheless, in the present study, RT was kept stable over the five temperatures assayed. Data on growth performance and plasma parameters in growing Iberian pigs (44 to 61 kg BW) at 30 °C have been previously published [3]. Contrary to the present study, RT increased at 30 compared to 20 °C. This finding was also reported in many studies with modern breeds. Li et al. [30] reported a modest increase in RT (less than 1 °C) during heat stress at 40 °C for 5 h, implying a heat tolerance in Bama miniature pigs (a non-improved breed like Iberian) in hot summer months. Small changes in RT when the temperature is elevated is a phenomenon reported before for animals adapt to hot environments [31] when water is available. Hao et al. [25] reported, in crossbred pigs under heat stress, an increase in water consumption to reduce the heat increment of feeding. Unfortunately, water consumption was not recorded in the present study due to a technical problem with the recording device. It seems that the Iberian pig could be adapted to heat in a similar way; a high body temperature means that the heat gain from the hot environment is reduced because the temperature gradient is reduced at the same time [31]. Another point to consider would be the adaptation of animals to maintain their HP—equivalent to losing more heat from the calorimetry point of view—when allocated into a hot environment, where their productive traits are affected but welfare would not be, probably due to the simple reason, the heat perception is less. Indirectly, this could have economic implications, as welfare leads to better health and resistance to diseases. This finding, which could be contradictory, indicates that Iberian pigs may be genetically adapted to tolerate much better elevated temperatures than modern breeds.

The constancy of RT and the small increase in ST implies that Iberian pigs have a mechanism to divert blood flow to more superficial areas to lose heat. In heat-stressed pigs, peripheral blood flow increases to promote non-evaporative heat dissipation [4], while blood flow to the internal organs decreases [32] as well as O_2_ consumption, and, thereafter, the HP. In pigs, the digestive tissues drained by the portal vein are responsible for 25% of total O_2_ consumption while their masses represent only 5% of total body mass [33]. In fact, the post-prandial total HP of Iberian pigs compared to Landrace gilts was greater (*p* < 0.05), whereas the HP of portal-drained viscera was lower (*p* < 0.05), together with a reduced post-prandial portal blood flow and a lower contribution of portal-drained viscera to the total HP [34], findings that can be advantageous in adapting to hot conditions.

During heat stress, thermoregulatory mechanisms contribute to promoting body heat loss. This involves an increase in pulmonary ventilation, BR and HR; therefore, heat stress is thought to increase the basal metabolic rate [35]. The NRC establishes for modern breeds a mean value of 338 and 413 kJ/kg^0.75^ BW and day (calculated on the basis of a 100 kg BW pig) for fasting HP—equivalent to net energy (NE) for maintenance, or basal metabolic rate or HP—and ME for maintenance, respectively, corresponding to a value of efficiency of the utilization of ME for maintenance (k_m_) of 0.8. There are no data about the net energy (NE) requirements for maintenance in Iberian pigs. Due to the high metabolic cost of the muscle protein turnover rate [11], fat pigs produce less heat per unit of metabolic size than lean pigs [36,37]. This suggests that pigs with a high potential for lean accretion may be more susceptible to heat stress. Nienaber et al. [38] reported a reduction of 4 °C in the UCT for pigs of newer genetics. In response to high ambient temperatures, Moreira et al. [39] reported that commercial pigs decreased their lean tissue deposition, whereas no effect was observed for Piau (non-improved breed) crossbred pigs, suggesting increased heat tolerance. Apart from that, commercial and Piau crossbred pigs had a similar magnitude of thermoregulatory activation in response to heat stress. These facts support the existence of genetic variation in the efficiency of energy use with respect to modern breeds. This is in agreement with the markedly increased basal HP with genetic selection for enhanced lean tissue accretion [40].

A way to reduce HP under heat stress would be to decrease visceral mass relative to BW. Interestingly, Pardo et al. [3] reported a numerical reduction in visceral mass relative to BW in Iberian pigs under heat stress conditions, although it did not attain statistical significance. Additionally, Le Bellego et al. [41] reported a decrease in total viscera weight in heat-stressed crossbred pigs. It is important to highlight that heat stress (33 vs. 23 °C [32]) may decrease viscera blood flow in pigs without noticeable changes in organ weight compared to pair-fed thermoneutral counterparts. Morales et al. [42] pointed out a reduction in the blood flow to the small intestines of heat-stressed pigs. As aforementioned, a reduction in portal blood flow was also reported when Iberian pigs were compared to Landrace pigs [34] in thermoneutral conditions.

The HR has seldom been used to determine the UCT. Patience et al. [43] reported in the Yorkshire × Landrace crossbreed an increase of 14 beats/min (*p* < 0.05) with a daily fluctuant temperature (from 20 to 38 °C). Similarly, Lykhach et al. [44], with a modern crossbreed, obtained an increase in HR at 28–31 °C with respect to pigs at thermoneutrality. However, HR remained stable as the temperature increased in the conditions of the present study. With this, O_2_ consumption was maintained, and therefore HP. It is well known that HR is an indirect estimator of the HP, on the basis of observed correlations between HR and O_2_ consumption, and consequently HP. Maybe, at very elevated temperatures, the balance between heat loss (evaporative and skin) and HP is positive when HR rises, but this could be the last resort before a heatstroke.

The ST increased with ambient temperature as a mechanism to increase heat loss. Pig skin shows good insulation due to the subcutaneous fat depth; however, there are metabolic mechanisms to divert fat deposition from external to internal sites [45]. It can be observed that the difference between RT and ST reaches a minimum coinciding with the marked negative effect of temperature on VFI that could be considered an indicator of the potential gradient of heat loss [46]. These authors [46] reported that ST was 2 and 4 °C below the RT from 18 to 29 °C, respectively. The RT was constant between 18 and 22 °C (38.6 °C), but it increased at higher temperatures (39.0 to 39.4 °C between 25 and 29 °C ambient temperature [46]). In the present study, ST was 1 °C lower than RT between 24 and 32 °C; the RT and ST difference was around 2.3 and 1.7 °C from 24 to 28 and 30 to 32 °C, respectively.

Quinou and Noblet [46] reported a linear increase in ST (0.29 °C) per degree of ambient temperature increased (between 18 and 25 °C), whereas the decrease in feed intake per degree was limited; at the same time, the BR also increased linearly and contributed to higher evaporative heat losses; above 25 °C, the ST and BR continued to increase. In the present study, both BR and ST also increased linearly but to a lesser extent (BR, 9.7 breaths/min per extra degree of ambient temperature (Equation (2)); and ST, 0.11 °C per extra degree (Equation (3))). Patience et al. [43] reported an increase in BR and RT of 90 breaths/min and 1.3 °C, respectively, between thermoneutral conditions (20 °C) and a pattern of elevated diurnal temperatures (from 20 to 38 °C within 24 h).

The reduction in RE when temperature rises [15,47] (Equation (5)) is more related to the decrease in VFI rather than the reduction in HP, as the correlation coefficients indicated (−0.74 vs. −0.53; Equations (1) and (4), respectively); certainly, VFI and HP were reduced by 40 and 16%, respectively, from 24 to 32 °C. Similar results were also obtained by other authors. This means that at the temperatures used, the efficiency (k_m_ and/or k_g_) values decreased to a lesser extent than expected for maintaining the RE. In fact, the gross efficiency of the utilization of ME (RE/ME intake) was maintained from 24 to 28 °C, decreasing (*p* < 0.01) at 30–32 °C. This may indicate that Iberian pigs under elevated temperature have the capacity to maintain their feed intake without increasing their HP, which could be an adaptation to the imposed elevated temperature. Rauw et al. [17] reported that increased ambient temperature resulted in reduced feed intake, whereas pigs (Duroc × Iberian) were able to maintain BW gains, but at lower temperatures than in the present study.

In general, selection for production under improved conditions leads to increased environmental sensitivity [48]. In an interesting study, Bloemhof et al. [49] reported differences in heat stress tolerance between two modern sow lines (purebred Yorkshire and purebred Large White) in their reproductive performance. At temperatures above the thermoneutral zone, these differences suggested that genetic selection for heat stress tolerance was possible. Sows of the line selected for reproductive performance in a temperate climate showed a reduction in this parameter when temperature increased. Selection regarding reproductive performance in the line from mainly tropical countries showed fewer problems with high temperatures. Therefore, the first line had greater reproductive performance under temperate conditions than the second, but the second was superior to the first when the outside temperatures exceeded 25 °C. This was clear proof of the genotype × environment interaction. Maybe Iberian pigs, as a non-improved breed, could be used in pig breeding programs for heat tolerance [12].

No change in RE does not necessarily imply unaltered RE partitioning to protein and fat. Protein deposition decreased in growing modern pigs reared at 30 compared with 23 °C [41], which is in accordance with increments in blood urea N in crossbred finishing pigs under heat stress (35 °C for 7 days) compared with pair-fed animals in thermoneutrality (20 °C for 7 days [50,51]). Pardo et al. [3] reported in Iberian pigs a tendency to increase fasting plasma urea N in heat stress compared to pair-fed pigs in thermoneutrality, maybe indicating increased protein catabolism. In addition, Pardo et al. [3] reported no changes in mesenteric fat as a percentage of empty BW, which is a proxy of visceral fat. Rinaldo and Le Dividich [45], in 30 kg BW Large White pigs, reported that between 25.0 and 31.5 °C, feed intake, RE and fat content decreased, whereas HP did not change; at 31.5 °C, protein deposition was less dependent on environmental temperature than fat retention. Similarly, Johnson et al. [52], in growing pigs, reported that heat stress reduced whole-body adipose accretion but had no effect on protein accretion rates. In the present study, this was supported by the fact that the RQ values were all above 1. The RQ value decreased (5.6% from 24 to 32 °C; *p* < 0.0001) as the temperature increased (Equation (6); *r* = −0.71). All RQ values indicated an overall lipogenesis which was less active from 30 °C onward. Indeed, de novo lipogenesis (Acetyl-CoA-carboxylase activity) in back fat, leaf fat and liver was less active in heat-stressed pigs compared to pair-fed thermoneutral counterparts [53]. Campos et al. [47] found a tendency to lower RQ values in pigs (65 kg BW; Piétrain × (Landrace × Large White)) at 30 °C than at thermoneutrality (1.025 vs. 1.050). Typically, low VFI produces a mobilization of adipose tissue and NEFA oxidation is markedly increased [54]. Sanz Fernandez et al. [55] reported that heat-stressed pigs tended to have reduced basal NEFA concentrations, suggesting that they did not mobilize as much adipose tissue as pigs in thermoneutrality. Reasons why heat-stressed animals fail to mobilize adipose tissue despite being in a hypercatabolic condition may be related to changes in insulin homeostasis. Normally, during insufficient nutrient intake, metabolic adaptations favor muscle growth at the expense of adipose tissue accretion [56,57,58]. However, when pigs and other animals are reared under heat stress and VFI decreases, carcasses typically deposit more fat than is energetically expected [59,60]. There is reduced lipolysis in adipose tissue as an attempt to reduce thermogenesis during mitochondrial fatty acid transport and ß-oxidation [61]. Heat-stressed pigs do not mobilize adipose tissue triglycerides but increase skeletal muscle proteolysis, minimizing lean tissue accretion and accumulating more lipids than bioenergetically expected [26]. Thus, heat-stressed animals have a limited ability to mobilize adipose tissue and, thus, are unable to maintain the mechanisms necessary to support the metabolic flexibility of fuel selection, displaying some postabsorptive metabolic changes that are in large part independent of reduced feed intake as adaptive mechanisms to prioritize the euthermia. An inability to mobilize adipose tissue reduces metabolic fuel options, allowing survival but reducing productivity [62].

In summary, the current study demonstrates that the UCT could be established for growing–finishing Iberian pigs at 30 °C for VFI (or ME intake), 28 °C for BR, 32 °C for ST, 32 °C for HP, 30 °C for RE and 32 °C for RQ. Nevertheless, HR and RT were not affected within the range of temperatures assessed (24–32 °C).

The Iberian pig is a very rustic breed which has been adapted to harsh environmental conditions for centuries, so it is not unexpected that as a result of adaptation to the environment, some physiological characteristics behave more favorably at high temperatures than in modern breeds.

According to Black et al. [63], the increase in the UCT can be around 15 °C between lactating sows and piglets; meanwhile no other study is available, the present data could be used to establish an approximate orientation of the UCT for other ages and sizes in the Iberian pig.

## 5. Conclusions

The physiological and metabolic responses of animals to heat is a very complex matter. Overall, in the present study, the upper critical temperature could be established at 28–30 °C in the growing–finishing Iberian pig, that is, 5 to 8 °C higher than the available data for modern breeds. However, these recommendations from this pioneering study carried out on Iberian pigs will need to be expanded with further investigations. In any case, the results indicate a good adaptation of Iberian pigs to hot environments.

## Figures and Tables

**Figure 1 animals-15-01374-f001:**
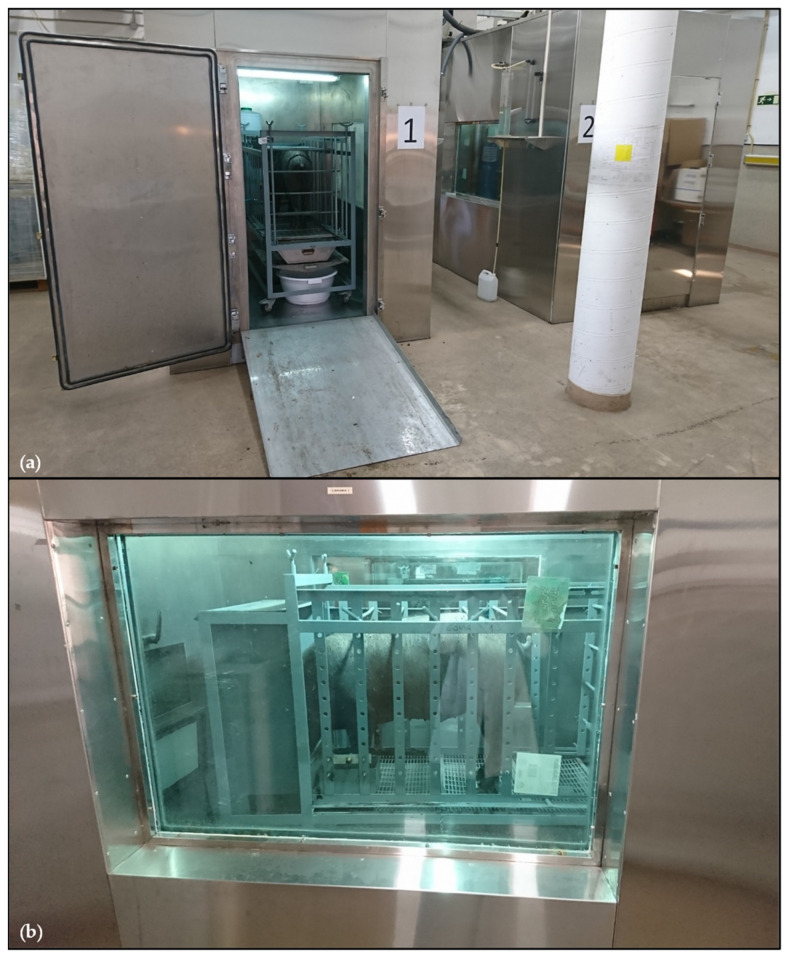
(**a**) Iberian pig on the slatted pen in the respirometry chamber; (**b**) image of the pig from the window of the chamber.

**Figure 2 animals-15-01374-f002:**
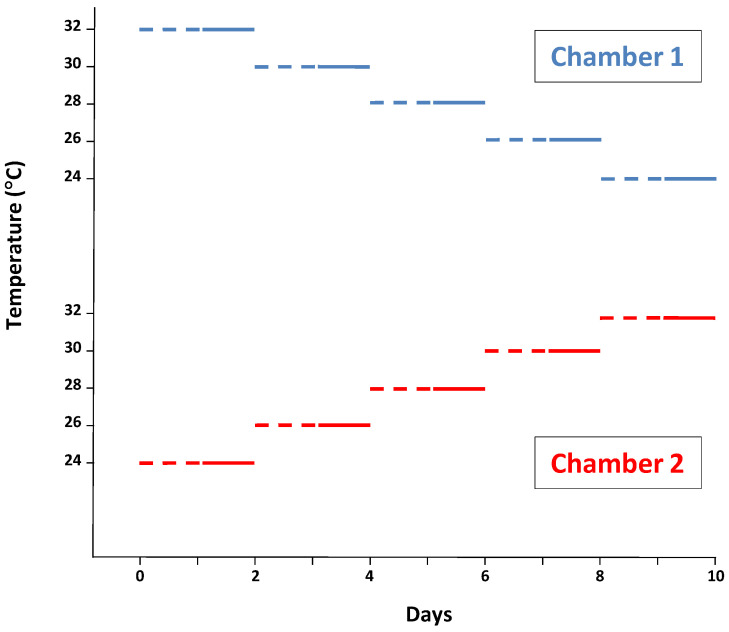
Cyclic variation in the temperature in the respirometry chambers over the 10 days of the experimental period. (- - -), adaptation period; (^____^), measurement period.

**Table 1 animals-15-01374-t001:** Composition and chemical analysis (g/kg as fed) of the control diet.

Ingredients	
Barley grain	359
Corn	150
Soft wheat	300
Soybean meal	100
Beet pulp	30
Lard	32
Monocalcium phosphate	2.0
Calcium carbonate	10
Sodium chloride	4.0
L-Lysine HCl (98%)	1.5
Vitamins and minerals ^1^	12
Chemical analysis	
Dry matter	905
Ash	48.0
Ether extract	49.9
Crude protein	136.1
Crude fiber	36
Metabolizable energy (MJ/kg) ^2^	13.1

^1^ Provided (per kg of diet): 2000 UI retinol as retinyl acetate, 800 UI cholecalciferol, 40 UI dL-α-tocopheryl acetate, 1.5 mg menadione, 2 mg thiamine, 3 mg riboflavin, 50 μg cyanocobalamin, 15 μg folic acid, 22.5 mg nicotinic acid, 15 mg d-pantothenic acid, 60 mg MnO, 80 mg FeCO_3_, 80 mg ZnO, 750 μg KI, 10 mg CuSO_4_ × 5H_2_O, 50 μg Na_2_SeO_3_, 250 mg sepiolite, 1.5 mg butylhydroxyanisole (BHA) and 7.5 mg butylhydroxytoluene (BHT). ^2^ From modified Atwater’s equation [13].

**Table 2 animals-15-01374-t002:** Average temperature (T; °C), relative humidity (RH; %) and temperature–humidity index (THI) in the respirometry chambers with Iberian pigs (*n* = 8) fed ad libitum a standard diet at 24, 26, 28, 30 and 32 °C temperature ^1^.

	24 °C	26 °C	28 °C	30 °C	32 °C	SEM	*p*-Value
T	23.9 ^a^	26.0 ^b^	27.9 ^c^	30.0 ^d^	31.9 ^e^	0.005	<0.001
RH	75.5 ^a^	77.1 ^b^	78.0 ^c^	77.5 ^bc^	75.3 ^a^	0.223	<0.001
THI	72.8 ^a^	76.1 ^b^	79.3 ^c^	82.4 ^d^	85.1 ^e^	0.031	<0.001

^1^ Calculated over 1056 data/temperature utilized. ^a–e^ Values within a row with different superscript letters significantly differ (*p* < 0.01).

**Table 3 animals-15-01374-t003:** Average body weight (BW; kg), voluntary feed intake (VFI; g DM/day), breathing rate (BR; breaths/min), heart rate (HR; beats/min), skin (ST; °C) and rectal (RT; °C) temperature, metabolizable energy (ME; kJ/kg^0.75^ BW and day) intake, heat production (HP; kJ/kg^0.75^ BW and day), retained energy (RE; kJ/kg^0.75^ BW and day) and respiratory quotient (RQ; CO_2_/O_2_) in Iberian pigs (*n* = 8) fed ad libitum a standard diet at 24, 26, 28, 30 and 32 °C temperature.

	24 °C	26 °C	28 °C	30 °C	32 °C	SEM	*p*-Value
BW	100.9	101.2	101.6	101.9	102.2	1.50	0.9763
VFI	3259.4 ^a^	2967.3 ^ab^	2783.6 ^ab^	2293.1 ^bc^	1946.2 ^c^	188.47	0.0001
BR	28.5 ^a^	55.0 ^ab^	74.0 ^bc^	99.1 ^c^	106.0 ^c^	8.89	<0.0001
HR	102.6	99.6	98.0	95.8	92.6	3.10	0.2225
ST	37.2 ^a^	37.4 ^ab^	37.5 ^ab^	38.0 ^ab^	38.1 ^b^	0.20	0.0146
RT	39.6	39.7	39.6	39.8	39.8	0.13	0.6023
ME intake	1477.4 ^a^	1341.2 ^ab^	1256.3 ^ab^	1033.6 ^bc^	872.5 ^c^	82.42	0.0001
HP	743.5 ^a^	769.2 ^a^	724.5 ^ab^	666.3 ^ab^	621.4 ^b^	28.35	0.0061
RE ^1^	734.0 ^a^	607.4 ^ab^	531.8 ^ab^	324.3 ^bc^	251.1 ^c^	68.37	0.0001
HP/ME intake	0.509 ^a^	0.554 ^ab^	0.592 ^ab^	0.662 ^b^	0.701 ^b^	0.037	0.0066
RE/ME intake	0.491 ^a^	0.446 ^ab^	0.408 ^ab^	0.338 ^b^	0.299 ^b^	0.037	0.0066
RQ	1.14 ^a^	1.13 ^a^	1.11 ^a^	1.08 ^ab^	1.02 ^b^	0.016	<0.0001

^1^ Calculated as RE = ME intake − HP. ^a–c^ Values within a row with different superscript letters were significantly different (*p* < 0.05).

## Data Availability

The data presented in this study are available on request from the corresponding author.

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
