# Peer review of "Upper Critical Temperature of Iberian Pigs"

_animals, 2025, doi:10.3390/ani15101374_

Round 1

Reviewer 1 Report

Comments and Suggestions for Authors

The manuscript addresses a relevant topic for which there is limited information available. The use of respirometry chambers in large animals provides scientifically valuable data. However, such studies are scarce because they require advanced and well-calibrated methodologies and are highly labor-intensive. Although the manuscript uses a limited number of animals, this is considered sufficient to obtain a general understanding of the metabolism of Iberian pigs under heat stress conditions.

A significant limitation in interpreting the results is that the animals were fed ad libitum, leading to the interference of two main effects: metabolic adaptation to heat stress and the direct effect of heat on feed intake. Addressing this issue would require a more complex experimental design, which is not feasible due to the methodological challenges inherent to this type of study. Nevertheless, the data obtained in this study are valuable because they reflect the overall impact of heat stress, encompassing all factors involved. This comprehensive approach is highly relevant in practical settings, as it mirrors the real-world conditions in productive environments and determines the biological response of animals to heat stress adaptation.

General comments.

An important concern is the statistical analysis of the results and the conclusions drawn, which in some cases appear premature and are not fully supported by the presented data. For instance, the graphs illustrating the response to ambient temperature suggest a linear response in all cases. This observation is incompatible with the concept of an inflection point or a change in trend beyond a specific temperature. However, the authors propose upper critical temperature (UCT) values without indicating how these were calculated or providing statistical significance for these results.

A reasonable alternative would be to determine whether the response is linear or quadratic in each case. If a quadratic response is identified, a standardized statistical procedure, such as the broken-line regression method, could be used to identify the point at which a change in trend occurs. This approach is commonly applied in similar studies evaluating dose-response relationships. Addressing this aspect is crucial and should be resolved in a revised version of the manuscript before publication.

From a general perspective, none charts in the graphs appear to show a double slope, suggesting that the response is linear across the studied temperature range. Consequently, the inflection point may lie outside this range.

Although not shown, the graphical representation of the respiratory quotient (RQ) does exhibit a response that appears to be both linear and quadratic, suggesting the existence of an inflection point around 29 °C. To substantiate this claim, rigorous statistical analysis would be required.

The RQ is likely the most scientifically robust parameter in this study, as it provides an external measure of metabolic changes in the animals. This parameter could be instrumental in proposing an inflection point that accounts for the dual effects of heat on metabolism and voluntary feed intake. Under heat stress, lipogenesis may be affected, as animals tend to reduce feed intake and prioritize mechanisms that decrease metabolic heat production. In such cases, the RQ may decrease due to lower carbohydrate utilization and increased fat oxidation. In this particular research, the RQ decreased as temperature increased, indicating reduced lipogenic activity at higher temperatures.

On the other hand, the SISCUSSION presented in the manuscript is overly general and fails to address critical aspects in depth. It would be advisable to delve into the discrepancies between the results obtained in this study and those reported in two recent publications by the same authors, which evaluated the effects of heat stress on Iberian pigs of different weights and reached entirely different conclusions. Simply stating that the results differ is insufficient; it is necessary to explore potential biological reasons for these differences and provide well-reasoned explanations or hypotheses.

Similarly, the effects of heat on voluntary FEED INTAKE and their comparison with other genotypes or existing data in the literature are not adequately discussed. Although the available values in the literature are heterogeneous, they should be referenced to establish comparisons and identify possible discrepancies, whether due to methodological differences or varying interpretations of the inflection point concept. For example, it is common to use the recommendations of Whittemore and Kyriazakis, who suggest that for every degree Celsius and kilogram of weight, feed intake decreases by approximately one gram under heat stress. An approximate calculation of the results obtained in this study suggests that Iberian pigs may experience a more pronounced reduction in feed intake compared to lean conventional pigs. This finding contrasts with previous results from the same research group, which indicated greater heat adaptation in rustic pigs.

It is possible, as suggested, that these conclusions were drawn prematurely. An alternative interpretation could be that rustic pigs significantly reduce their feed intake to maintain a lower metabolic rate, which might serve as an adaptive mechanism to heat stress. This behavior aligns with their higher subcutaneous fat proportion and greater endogenous metabolic workload, characteristics previously described by the authors. However, this hypothesis requires more robust statistical underpinning and detailed analysis.

Finally, the authors mention high variability in voluntary feed intake under heat stress but do not provide concrete results or comparisons with data from other studies involving lean pigs. Analyzing the coefficient of variation of feed intake in relation to temperature and comparing it across genotypes would be highly valuable for drawing scientifically valid conclusions rather than relying on general comments unsupported by numerical data.

In conclusion, this is an interesting and technically well-conducted study, but it requires substantial revision. The statistical analysis should be improved to adequately support the study's claims and proposals. Additionally, the discussion needs to be enriched with more detailed analysis and critical comparisons with relevant previous studies. This will ensure that the conclusions are robust and scientifically well-founded.

Specific comments.

Tables and figures should include the number of experimental units in each case (n = ..).

Line 11. Text: "the heat production reach a minimum...". Use reaches better

Line 28. Text: "The UCT can be stablished in 28 for BR..." Consider "established."
Line 42. Consider: "...where pigs are free on the land and balanced mixed diets are offered. This is followed by a finishing phase..."

Line 145. identical to before.

Line 162. "...the chamber was opened for cleaning and providing feed to the pigs."

Line 251. "Linear regressions were established..."

Line 472. “with”

Line 389 "temperature."

Line 472 "...which is 5 to 8 °C higher than the available data for m

Author Response

Reviewer 1

Comments and Suggestions for Authors

The manuscript addresses a relevant topic for which there is limited information available. The use of respirometry chambers in large animals provides scientifically valuable data. However, such studies are scarce because they require advanced and well-calibrated methodologies and are highly labor-intensive. Although the manuscript uses a limited number of animals, this is considered sufficient to obtain a general understanding of the metabolism of Iberian pigs under heat stress conditions.

A significant limitation in interpreting the results is that the animals were fed ad libitum, leading to the interference of two main effects: metabolic adaptation to heat stress and the direct effect of heat on feed intake. Addressing this issue would require a more complex experimental design, which is not feasible due to the methodological challenges inherent to this type of study. Nevertheless, the data obtained in this study are valuable because they reflect the overall impact of heat stress, encompassing all factors involved. This comprehensive approach is highly relevant in practical settings, as it mirrors the real-world conditions in productive environments and determines the biological response of animals to heat stress adaptation.

General comments.

An important concern is the statistical analysis of the results and the conclusions drawn, which in some cases appear premature and are not fully supported by the presented data. For instance, the graphs illustrating the response to ambient temperature suggest a linear response in all cases. This observation is incompatible with the concept of an inflection point or a change in trend beyond a specific temperature. However, the authors propose upper critical temperature (UCT) values without indicating how these were calculated or providing statistical significance for these results.

A reasonable alternative would be to determine whether the response is linear or quadratic in each case. If a quadratic response is identified, a standardized statistical procedure, such as the broken-line regression method, could be used to identify the point at which a change in trend occurs. This approach is commonly applied in similar studies evaluating dose-response relationships. Addressing this aspect is crucial and should be resolved in a revised version of the manuscript before publication. Reviewer is right. We have performed quadratic polynomial regression equations, but the quadratic polynomial terms were not significant. According to the equations, plateaus are established in all cases outside the temperatures studied and in some cases with no biological meaning. For example, for retained energy the plateau would be reached at -28.69 ºC; for breathing rate, it would be reached at 36.42 ºC when the experience in our previous works and also with a direct observation in laboratory or commercial Iberian pig farms, the iberian pigs increased their rate about 27-29 ºC; for voluntary food intake the point where intake would decrease it would be located at a temperature of 16.74 ºC, a relatively low temperature at which pigs should theoretically ingest more energy to increase their heat production and thus maintain their body temperature. Everything indicates that in order to improve the equations, lower and/or higher temperatures should have been included depending on the parameter studied, although this would have made the design extremely complicated, even from the point of view of animal welfare. According to the data appearing in Table 3, the real temperatures at which a change in parameters is observed would be between 26-28 ºC, on average. Although all the parameters studied add useful information, the truly important ones are voluntary food intake, heat production and the relationship between both (retained energy (also the inefficiency and efficiency values)) where the decrease (P<0.05) in all of them is between 28-30ºC. Tukey’s multiple-range test was used to establish the temperature where a shift was produced in the value of the parameter studied and establish such temperature as the UCT because only 5 temperatures were used in the range of the expected UCT. The linear equations were adjusted in a similar way to the polynomial ones, so we chose them for their simplicity when it comes to being applied from a practical point of view into a similar temperature range. Although these equations add some extra information. if the reviewer considers appropriate, we can delete them.

From a general perspective, none charts in the graphs appear to show a double slope, suggesting that the response is linear across the studied temperature range. Consequently, the inflection point may lie outside this range. Figures were added to give a visual information to readers but are empty of statistical information. As it was aforementioned, in order to improve the figures, lower and/or higher temperatures may have been included depending on the parameter studied, although this would have increase extremely the experiment duration, complicating the permits from the bioethics committees. Although the number of temperatures may be questioned, the fact that we only have two respirometry chambers in addition that the animal growth up quickly, it makes difficult to increase the number of temperatures (or animals). Additionally, from an animal welfare standpoint (three Rs principle: Reduction), we maintained the required number of animals to a minimum as requested by the Bioethics Committee and the time confined on the slatted pen. We had to reach a compromise between number of temperatures (or animals) and can get the Bioethical approval.

Although not shown, the graphical representation of the respiratory quotient (RQ) does exhibit a response that appears to be both linear and quadratic, suggesting the existence of an inflection point around 29 °C. To substantiate this claim, rigorous statistical analysis would be required. We have also made a polynomial equation for the RQ and, as with the others parameters, it would reach its plateau outside the temperatures studied (23 ºC). What indicates RQ in Table 3 is that lipogenesis is affected from 30 ºC but pigs did not use their energy reserves as metabolic fuel. RQ is used as complement for the real important parameters that define the upper critical temperature (feed intake and, mainly, heat production).

The RQ is likely the most scientifically robust parameter in this study, as it provides an external measure of metabolic changes in the animals. This parameter could be instrumental in proposing an inflection point that accounts for the dual effects of heat on metabolism and voluntary feed intake. Under heat stress, lipogenesis may be affected, as animals tend to reduce feed intake and prioritize mechanisms that decrease metabolic heat production. In such cases, the RQ may decrease due to lower carbohydrate utilization and increased fat oxidation. In this particular research, the RQ decreased as temperature increased, indicating reduced lipogenic activity at higher temperatures. Right. However, even when feed intake experimented a reduction when the ambient temperature was increased, Iberian pig is a fatty pig and capable to maintain the lipogenesis as indicated the RQ values where in all the cases it was over 1.

On the other hand, the SISCUSSION presented in the manuscript is overly general and fails to address critical aspects in depth. It would be advisable to delve into the discrepancies between the results obtained in this study and those reported in two recent publications by the same authors, which evaluated the effects of heat stress on Iberian pigs of different weights and reached entirely different conclusions. Simply stating that the results differ is insufficient; it is necessary to explore potential biological reasons for these differences and provide well-reasoned explanations or hypotheses. In the previous studies reported by us the objective was very different. Then, the pigs were under 30 ºC for 28 days, that is, under long-term constant heat stress conditions, with almost half of the BW and with the aim to determinate the effect of dietary betaine and zinc on growth. In addition, in the present study, pigs were in a finishing phase -montanera-, and 2 days in each of the temperatures assayed. Then, it is difficult to extrapolate conclussions from those studies to the present one, also because in those studies only two temperatures were assayed: 20 and 30 ºC. In fact, thehe lack of knowledge of whether 30°C was stressful for Iberian pigs made us consider carrying out this study. No difference (P>0.05) was obtained between the five treatments studied for heat production but the difference between pigs at 20 and 30 ºC was numerically much higher (184 kJ/kg0.75 BW instead of 78 kJ/kg0.75 BW in the present study) between the comparable temperature treatments (24 and 30 ºC). A sentence has been added (L302-306: Iberian pig ME intake (measured during one day into a respirometry chamber under long-term constant heat stress conditions) was not reduced (p > 0.1) at 30 °C and, therefore, growth (RE) was not affected, which may be due to a genetic adaptation to harsh environments [8]. However, Pardo et al. [3] reported a reduction in VFI obtained along 28 experimental days in growing Iberian pig under long-term heat) to clarify this fact.

Similarly, the effects of heat on voluntary FEED INTAKE and their comparison with other genotypes or existing data in the literature are not adequately discussed. Although the available values in the literature are heterogeneous, they should be referenced to establish comparisons and identify possible discrepancies, whether due to methodological differences or varying interpretations of the inflection point concept. For example, it is common to use the recommendations of Whittemore and Kyriazakis, who suggest that for every degree Celsius and kilogram of weight, feed intake decreases by approximately one gram under heat stress. An approximate calculation of the results obtained in this study suggests that Iberian pigs may experience a more pronounced reduction in feed intake compared to lean conventional pigs. This finding contrasts with previous results from the same research group, which indicated greater heat adaptation in rustic pigs. Part of the reasoning was explained in our earlier comments. It is not a surprise that Iberian behaves different. The intake capacity of Iberian is much greater than the in comparison to leaner breeds (Morales et al., 2002. Comparative digestibility and lipogenic activity in Landrace and Iberian finishing pigs fed ad libitum corn- and corn–sorghum-acorn-based diets. Livest. Prod. Sci. 77, 195–205; Nieto et al., 2012. Response analysis of the Iberian pig growing from birth to 150 kg body weight to changes in protein and energy supply. J. Anim. Sci. 90, 3809-3820) then the range for reduction of intake per degree of temperature is proportionally higher than for leaner breeds.

It is possible, as suggested, that these conclusions were drawn prematurely. An alternative interpretation could be that rustic pigs significantly reduce their feed intake to maintain a lower metabolic rate, which might serve as an adaptive mechanism to heat stress. This behavior aligns with their higher subcutaneous fat proportion and greater endogenous metabolic workload, characteristics previously described by the authors. However, this hypothesis requires more robust statistical underpinning and detailed analysis. The design of the study was based in direct observations of the Iberian pigs because no data about the UCT of rustic pigs was published in literature. We give some possible reasons about why the Iberian shows differences respect with the modern breeds. Reviewer is right about his appreciations and we have the intention to keep going this kind of studies. Form the data we have reported there is a change in most of parameters studied at around 28-30 ºC and further investigation is warranted.

Finally, the authors mention high variability in voluntary feed intake under heat stress but do not provide concrete results or comparisons with data from other studies involving lean pigs. Analyzing the coefficient of variation of feed intake in relation to temperature and comparing it across genotypes would be highly valuable for drawing scientifically valid conclusions rather than relying on general comments unsupported by numerical data. The amplitude of data about lean pigs makes to establish comparison very difficult, moreover, Iberian voluntary feed intake is very high in comparison with lean breeds (please see the references mentioned above). This is a good idea for a review paper but we believe that is out of the scope of the present paper.

In conclusion, this is an interesting and technically well-conducted study, but it requires substantial revision. The statistical analysis should be improved to adequately support the study's claims and proposals. Additionally, the discussion needs to be enriched with more detailed analysis and critical comparisons with relevant previous studies. This will ensure that the conclusions are robust and scientifically well-founded. Reviewer is right, however as it is the first manuscript that determine the upper critical temperature (UCT) in a non-improved breed there is a lack of information. We have utilized the available information about the UCT but the data refer to modern breeds.

Specific comments.

Tables and figures should include the number of experimental units in each case (n = ..). It already was included.

Line 11. Text: "the heat production reach a minimum...". Use reaches better OK

Line 28. Text: "The UCT can be stablished in 28 for BR..." Consider "established." OK
Line 42. Consider: "...where pigs are free on the land and balanced mixed diets are offered. This is followed by a finishing phase..." OK

Line 145. identical to before. OK

Line 162. "...the chamber was opened for cleaning and providing feed to the pigs." OK

Line 251. "Linear regressions were established..." OK

Line 472. “with” OK

Line 389 "temperature." OK

Line 472 "...which is 5 to 8 °C higher than the available data for m OK

Reviewer 2 Report

Comments and Suggestions for Authors

The study focused on the Upper Critical Temperature of Iberian Pig addresses a highly relevant and timely topic, particularly in light of climate change and its implications for animal welfare and productivity in livestock systems. The findings offer valuable insights into the thermoregulatory adaptations of Iberian pigs, a breed known for its resilience in challenging environments. However, there are several weaknesses that need to be addressed. The small sample size (n = 8) limits the generalizability of the results, and the absence of water consumption data due to technical issues represents a significant gap, as water intake is a critical factor in heat stress management. Furthermore, the discussion would benefit from a deeper exploration of the mechanisms underlying key findings, such as the stability of rectal temperature and the trends in heat production and respiratory quotient. Addressing these limitations and expanding the dataset in future studies would significantly enhance the robustness and impact of your research.

Author Response

Reviewer 2

Comments and Suggestions for Authors

The study focused on the Upper Critical Temperature of Iberian Pig addresses a highly relevant and timely topic, particularly in light of climate change and its implications for animal welfare and productivity in livestock systems. The findings offer valuable insights into the thermoregulatory adaptations of Iberian pigs, a breed known for its resilience in challenging environments. However, there are several weaknesses that need to be addressed. The small sample size (n = 8) limits the generalizability of the results, and the absence of water consumption data due to technical issues represents a significant gap, as water intake is a critical factor in heat stress management. Furthermore, the discussion would benefit from a deeper exploration of the mechanisms underlying key findings, such as the stability of rectal temperature and the trends in heat production and respiratory quotient. Addressing these limitations and expanding the dataset in future studies would significantly enhance the robustness and impact of your research.

Thak you for your comments.

Comments and answers to the Reviewer have been included into the “pdf” document.

The new sentences added to the manuscript are into the “doc” document using the “track changes” tool.

Reviewer 3 Report

Comments and Suggestions for Authors

The design of this study is rigorous, however, the 24°C breathing rate has been high, because the test method is correct, the results are reliable, but the inference of UCT 30°C, the subjective component is too much, the reviewer has defined the critical body temperature of pigs as 39.5°C, RT 39.6°C at 24°C in this paper. The results of this paper show a decrease in intake and HP, and an increase in fat deposition is inferred, further illustrating that UCT cannot be inferred with an increase in HP. Because the results are reliable and the inferences are well-founded, they can be published for researchers' reference.

Author Response

Reviewer 3

Comments and Suggestions for Authors

The design of this study is rigorous, however, the 24°C breathing rate has been high, because the test method is correct, the results are reliable, but the inference of UCT 30°C, the subjective component is too much, the reviewer has defined the critical body temperature of pigs as 39.5°C, RT 39.6°C at 24°C in this paper. The results of this paper show a decrease in intake and HP, and an increase in fat deposition is inferred, further illustrating that UCT cannot be inferred with an increase in HP. Because the results are reliable and the inferences are well-founded, they can be published for researchers' reference.

Thank you.

Round 2

Reviewer 1 Report

Comments and Suggestions for Authors

Thank you for your comments and responses.
I am pleased to learn that you have applied quadratic polynomial regression equations in your analysis. The results appear to be contrary to expectations, as the quadratic terms were not statistically significant and the plateaus were observed outside the tested temperature ranges in all cases. This is an interesting finding. I encourage you to include this methodology in the Materials and Methods section, as well as in the Results and Discussion. Potential explanations for these findings would be of interest to readers and could contribute to a better understanding of the complex phenomena involved. It is important to note that the absence of an expected result still constitutes a valuable result.

In light of this, I believe that the statement in lines 476–479 — “the upper critical temperature is established at 28–30 °C in the growing-finishing Iberian pig, which is 5 to 8 °C higher than the available data for modern breeds” — is not adequately supported by the presented data. I regret to say that I do not observe any clear inflection point in Table 3 or in Figures 3–5 that would substantiate this claim.

Overall, I find the manuscript to be of interest, the methodology to be sound, and I believe it merits publication. However, I do not  recommend publication in its present form and euncorage authors to present conclusions closely aligned with the data and supported by statistical evidence.

Author Response

Reviewer 1 (Round 2)

Comments and Suggestions for Authors

Thank you for your comments and responses.

I am pleased to learn that you have applied quadratic polynomial regression equations in your analysis. The results appear to be contrary to expectations, as the quadratic terms were not statistically significant and the plateaus were observed outside the tested temperature ranges in all cases. This is an interesting finding. I encourage you to include this methodology in the Materials and Methods section, as well as in the Results and Discussion. Potential explanations for these findings would be of interest to readers and could contribute to a better understanding of the complex phenomena involved. It is important to note that the absence of an expected result still constitutes a valuable result. Thank you for your interesting points. A sentence about this subject have been included in the text (L276-289: “In the present study quadratic polynomial regression equations were performed, but the quadratic polynomial terms were not significant. According to the equations, the different plateaus (estimated UCT) were established in all cases outside the range of temperatures studied and in some cases with no biological meaning. For example, for RE the plateau would be reached at -28.7 ºC; for BR at 36.4 ºC, in contrast with our experience in previous works and also with direct observations in laboratory or commercial pig farms, when the Iberian pigs increased their rate at about 27-29 ºC; for VFI the plateau was at 16.7 ºC, a temperature at which pigs should theoretically ingest more energy to increase their HP to maintain the body temperature. These evidences indicating that lower and/or higher temperatures, depending on the parameter studied, should have been included in the experimental design, although this would have made it extremely complicated, also with issues of animal welfare. The linear equations were chosen for their simplicity when it comes to being applied from a practical point of view into a similar temperature range.”).

In light of this, I believe that the statement in lines 476–479 — “the upper critical temperature is established at 28–30 °C in the growing-finishing Iberian pig, which is 5 to 8 °C higher than the available data for modern breeds” — is not adequately supported by the presented data. I regret to say that I do not observe any clear inflection point in Table 3 or in Figures 3–5 that would substantiate this claim. The sentence has been modified to be less blunt (L490-494: “In the present study, the upper critical temperature could be established at 28-30 °C in the growing-finishing Iberian pig, that is, 5 to 8 °C higher than the available data for modern breeds. However, these recommendations from this pioneering study carried out on Iberian pigs will need to be expanded with further investigation. In any case, the results indicate a good adaptation of Iberian pigs to hot environments”.).

Overall, I find the manuscript to be of interest, the methodology to be sound, and I believe it merits publication. However, I do not  recommend publication in its present form and euncorage authors to present conclusions closely aligned with the data and supported by statistical evidence. OK, we have slightly changed the conclusions to address this issue.

Reviewer 2 Report

Comments and Suggestions for Authors

The study on the Upper Critical Temperature of Iberian Pigs explores a crucial topic with significant implications for livestock management under climate change. The findings provide valuable insights into the thermoregulatory adaptations of this resilient breed. However, the dataset presented in this article is insufficient to offer a comprehensive understanding of the subject.

Author Response

Reviewer 2 (Round 2)

Comments and Suggestions for Authors

The study on the Upper Critical Temperature of Iberian Pigs explores a crucial topic with significant implications for livestock management under climate change. The findings provide valuable insights into the thermoregulatory adaptations of this resilient breed. However, the dataset presented in this article is insufficient to offer a comprehensive understanding of the subject.

Thank you very much for your time and effort. We are sorry that our responses did not satisfy your comments and questions. We accept your opinion, but we do not share it. The range of five temperatures to study its effect on different pig parameters has been consistently used (e.g. Quiniou and Noblet. (1999), J. Anim. Sci.; Quiniou, et al. 2001, Br. J, Nutr.; etc.).

Our study is the first study on upper critical temperature of Iberian pigs and uses all the usual physiological parameters, plus other possible ones that are rarely used (heat production) in institutes and laboratories due to the necessary equipment and technical training. It also includes other associated parameters that, in our view, add value, such as respiratory quotient (RQ) and retained energy. The latter three parameters have been measured using calorimetric techniques.

Anyway, we reiterate our gratitude for your time and effort to improve our manuscript.